# Block-Active ADMM to Minimize NMF with Bregman Divergences

**DOI:** 10.3390/s23167229

**Published:** 2023-08-17

**Authors:** Xinyao Li, Akhilesh Tyagi

**Affiliations:** Department of Electrical and Computer Engineering, Iowa State University, Ames, IA 50010, USA; xli@iastate.edu

**Keywords:** NMF, ADMM, Bregman divergence, block active, imaging sensor

## Abstract

Over the last ten years, there has been a significant interest in employing *nonnegative matrix factorization* (NMF) to reduce dimensionality to enable a more efficient clustering analysis in machine learning. This technique has been applied in various image processing applications within the fields of computer vision and sensor-based systems. Many algorithms exist to solve the NMF problem. Among these algorithms, the *alternating direction method of multipliers* (ADMM) and its variants are one of the most popular methods used in practice. In this paper, we propose a block-active ADMM method to minimize the NMF problem with general Bregman divergences. The subproblems in the ADMM are solved iteratively by a *block-coordinate-descent-type* (BCD-type) method. In particular, each block is chosen directly based on the *stationary condition*. As a result, we are able to use much fewer auxiliary variables and the proposed algorithm converges faster than the previously proposed algorithms. From the theoretical point of view, the proposed algorithm is proved to converge to a stationary point sublinearly. We also conduct a series of numerical experiments to demonstrate the superiority of the proposed algorithm.

## 1. Introduction

### 1.1. Overview of the Matrix Factorization Algorithms

Unsupervised learning is a form of machine learning in which models are trained on unlabeled data to classify patterns or make inferences without any external guidance or supervision. The key advantage of unsupervised learning is its ability to uncover hidden structures and relationships within datasets. This may not be readily apparent through manual inspection, enabling the automatic discovery of insights and patterns in large datasets. However, working with large datasets can be challenging because they are often noisy and high-dimensional, which makes their processing and analysis difficult. To address this challenge, researchers often use dimensionality reduction techniques to extract meaningful features from the data. By reducing the dimensionality of the data, the computation cost of training and classification algorithms can be improved. Dimensionality reduction can be achieved by reducing the size of the feature vector, which is the input data used to train and test machine learning models. Unsupervised learning methods factor the data matrix subject to various constraints for such dimensionality reduction. Depending on the constraints, the resulting factors have significantly different data representations. *Principal component analysis* (PCA) [1] enforces no constraints on the factorization. Consequently, PCA achieves an optimal low-dimension approximation to the data matrix while retaining as much of the original variation as possible. For this reason, PCA has been widely applied in various applications such as face recognition [2,3,4] and document representation [5,6,7].

In parallel, previous studies have shown that there is some psychological and physiological evidence for parts-based representation in the human brain [8,9,10]. *Nonnegative matrix factorization* (NMF) [8] has been proposed to learn the parts of objects by enforcing the nonnegative constraints. In particular, NMF approximates the data matrix by a product of two nonnegative matrices. The nonnegative constraints are useful to learn a parts-based representation of the data because it only allows additive combinations. It has been shown nonnegative matrix factorization is superior to PCA in fields that use nonnegative datasets such as face recognition [11,12,13,14], document clustering [15,16], audio signal processing [17,18], and recommendation systems [19,20].

### 1.2. Nonnegative Matrix Factorization

*Nonnegative Matrix Factorization* (NMF) is a technique for factorizing a matrix into two nonnegative matrices, denoted as *W* and *H*. This method is distinct because it constrains all elements in *W* and *H* to be nonnegative. To grasp the concept of NMF, it is essential to comprehend the underlying intuition behind matrix factorization.

As shown in Figure 1, suppose we have a matrix *V* of size m×n, where each element is greater than or equal to zero. With NMF, we can decompose *V* into two matrices: *W* of size m×k and *H* of size k×n, where *k* is a chosen rank. Notably, both *W* and *H* have only nonnegative elements. Here, *V* is defined as:(1)$$\begin{array}{c} V_{m\times n}=W_{m\times k}H_{k\times n} \end{array}$$
where
*V* is the original input matrix (Linear combination of *W* and *H*);*W* is the feature matrix;*H* is the coefficient matrix;*k* is the low-rank approximation of *V* (k≤min(m,n)).

**Figure 1 sensors-23-07229-f001:**
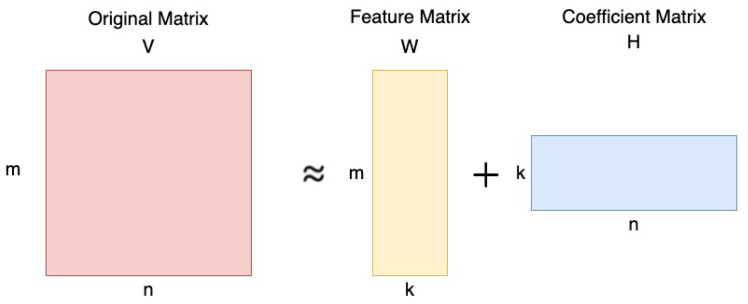
NMF intuition.

The primary goal of NMF is to perform dimensional reduction and feature extraction. By specifying a lower dimension *k*, the main objective of NMF is to identify two matrices, $W\in R_{m\times k}$ and $H\in R_{k\times n}$, containing only nonnegative elements, as illustrated in Figure 1.

Specifically, by setting k≤min(m,n), the factorization process breaks down the original matrix *V* into two matrices. Therefore, the dimensional reduction occurs as the original matrix *V* of size m×n is represented as the product of a smaller matrix *W* of size m×k and a smaller matrix *H* of size k×n, resulting in a dimensional reduction from m×n to m×k+k×n. Note that (m×k+k×n)≤(m×n) since k≤min(m,n). Typical machine learning algorithms, including statistical machine learning and deep learning methods such as convolutional neural networks (CNN), take a time superlinear in the input size. This training time and classification time also depend heavily on the feature space size. NMF likely reduces both the input size and the feature vector size. With the training time and classification time being superlinear, the efficiency of both improves significantly with NMF.

The underlying assumption of NMF is that the input comprises a set of latent features, each of which is represented by a column in the *W* matrix. Moreover, each column in the *H* matrix represents the “coordinates of a data point” in the *W* matrix, essentially holding the weights related to matrix *W*. In essence, each data point represented by a column in *V* can be approximated by a summation of nonnegative vectors represented by columns in *W* weighted by a row in *H*.

#### 1.2.1. Image Processing—Facial Feature Extraction

In order to better understand the intuition behind the NMF algorithm, we consider real-world scenarios, specifically the application of the algorithm to image processing. Suppose we have an input image consisting of pixels that form matrix *X*. NMF produces two factors (W,H) such that each image X(:,j) is approximated as a linear combination of the columns in *W*. As shown in Figure 2, for facial images, the columns of *W* can be interpreted as basic images consisting of features such as eyes, noses, mustaches, and lips. The columns of *H* indicate the presence of these features in the corresponding *X* image.

#### 1.2.2. Contributions

This paper makes the following innovative contributions revolving around a novel algorithm for tackling NMF challenges:We present a coordinate descent approach coupled with an innovative strategy for selecting coordinates to address the ADMM subproblems.In contrast to the classic ADMM and multiplicative update methods, our proposed algorithm attains a notably reduced error level while showcasing enhanced convergence characteristics, marked by an enhanced stability, smoother trajectories, and expedited convergence.We establish the effectiveness of our approach through a rigorous theoretical analysis and substantiate our claims via an array of comprehensive experiments conducted on synthetic and real datasets in Section 6. These experiments collectively serve to underscore the superior performance and potential of our novel methodology.

#### 1.2.3. Discussion

Comparing the proposed method in Algorithm 3 with the classical ADMM, we use much fewer primal and dual decision variables. Specifically, the ADMM in Algorithm 1 introduces new primal variables $W_{+}$ and $H_{+}$, and dual variables $\alpha_{X}$, $\alpha_{W}$, and $\alpha_{H}$, while the proposed method in Algorithm 3 introduces no primal variables, and only one dual variable $\alpha_{X}$. This helps with the efficiency of the algorithm.In Section 4, we introduce a new approach termed the “block active method” designed to tackle the problems formulated in (Equation 14). Our central result, as established in Theorem 4, rigorously demonstrates that under reasonable assumptions, our proposed method converges towards a stationary point denoted as $x^{*}$ in Equation (Equation 15) at a sublinear rate of convergence.To expound on this, we demonstrate that the error, as defined by $f\left(x^{k}\right)-f\left(x^{*}\right)$ on the left-hand side of the equation within Theorem 4, consistently diminishes. This reduction is characterized by the relation $f\left(x^{k}\right)-f(x)=O\left(k^{-1}\right)$, indicative of the error’s gradual decline to zero with iteration count *k* approaching infinity. This type of convergence behavior is denoted as sublinear [22] due to its property of diminishing error reduction over iterations. This stands in contrast to the linear convergence typified by expressions such as $\gamma^{k}$ for some constant 0<γ<1, where the decline in error remains consistent.NMF finds applications in tasks such as face recognition, document clustering, audio signal processing, and recommendation systems. When employing NMF to address analogous optimization problems, there should not be any difference in the theoretical results.The image resolution may or may not affect the results. Given NMF is a nonconvex optimization problem, a global min cannot be guaranteed to be found in the general setup. The quality of the solution a method converges to depends on several factors, such as the initialization of *W*, *H*, and *X*, and the learning rate rho. Thus, improving the resolution of the data or quality of the data may or may not improve the result.

#### 1.2.4. Paper Organization

This paper is organized as follows. Section 2 introduces the NMF problem and places it within the context of the related existing research. Section 3 provides a concise overview of the widely recognized ADMM. Section 4 presents the proposed *block-active method* as a coordinate descent approach coupled with an innovative strategy for selecting coordinates to address the ADMM subproblems. Section 5 introduces an innovative ADMM-style approach for addressing the NMF problem (Equation 2). Section 6 assesses the experimental outcomes across both synthetic and real datasets, while Section 7 serves as the concluding segment.

## 2. NMF Problem and Previous Work

The NMF problem can be formulated as follows:(2)$$\begin{array}{cc} \min & \;f(W,H)\equiv D(V|WH), \end{array}$$(3)$$\begin{array}{cc} s.t. & \;W_{ik},H_{kj}\geq 0, \end{array}$$
where $D(V|\hat{V})$ represents some measure of divergence between *V* and its approximation $\hat{V}$. A general family of divergence functions is the β-divergence, denoted by $D_{\beta }$. The β-divergence between two matrices is defined as the sum of the elementwise divergence, i.e., $D_{\beta }\left(V,\hat{V}\right)$ = $\sum_{i,j}d_{\beta }\left(V_{ij}|{\hat{V}}_{ij}\right)$, where $d_{\beta }$ is defined by
(4)$$\begin{array}{c} d_{\beta }\left(x|y\right)=\frac{x^{\beta }}{\beta (\beta -1)}+\frac{y^{\beta }}{\beta }-\frac{xy^{\beta -1}}{\beta -1}. \end{array}$$

The three most commonly used β-divergence functions with NMF in practice are the Euclidean distance, Kullback–Leibler (KL) divergence, and Itakura–Saito (IS) divergence so as to model the Gaussian noise, Poisson noise, and multiplicative gamma noise, respectively. Particularly, we have

β=2 (Euclidean distance): $d\left(x|y\right)=\frac{1}{2}{\left(x-y\right)}^{2}$;β=1 (Kullback–Leibler divergence): $d\left(x|y\right)=x\log\frac{x}{y}-x+y$;β=0 (Itakura–Saito divergence): $d\left(x|y\right)=-\log\frac{x}{y}+\frac{x}{y}-1$.

In the literature on NMF, many algorithms have been proposed to solve Problem (Equation 2) for β=2, including *multiplicative updates* (MU) [23,24,25,26], projected gradient descent (PGD) [27], hierarchical alternating least square (HALS) [28,29,30], the alternating direction method of multipliers (ADMM) [31,32], and alternating nonnegative least square (ANLS) [33]. Unfortunately, there are few works proposed to solve the NMF problem with the general Bregman divergence. In this paper, we propose an ADMM that can be used to solve the NMF problem with the general Bregman divergence. In fact, we are not the first one proposing an ADMM method to solve the NMF problem. For example, reference [31] also proposed an ADMM method to solve NMF problem and each subproblem had a closed-form solution. However, our method introduces a much fewer number of auxiliary variables, and we use an iterative method to solve each subproblem. By doing so, the proposed algorithm converges much faster than the previously proposed algorithms. We provide the theoretical analysis of the proposed algorithm and construct a line of numerical experiments to demonstrate the performance of the proposed method.

## 3. Alternating Direction Method of Multipliers

In this section, we provide a brief review of the well-known *alternating direction method of multipliers* (ADMM). We consider an optimization problem formulated as follows:(5)$$\begin{array}{c} \min\;f(x)\;s.t.\;Ax=b \end{array}$$
where $f:R^{n}\to R$ is the objective function, $x\in R^{n}$ is the decision variable, $A\in R^{m\times n}$ and $b\in R^{m}$ are a given matrix and vector. Since the objective function *f* could be nonconvex, searching for a global solution is not easy. Instead, the common pursuit is to find a *stationary point* of the problem. A stationary point of (Equation 5) is a vector $x^{*}$ that satisfies
(6)$$\begin{array}{cc} \nabla f\left(x^{*}\right)+A^{T}y & =0 \end{array}$$
(7)$$\begin{array}{cc} Ax^{*} & =b \end{array}$$

The ADMM method can help us find such vector $x^{*}$. In particular, the *augmented Lagrangian function* is given by
$$\begin{array}{c} L_{\rho }\left(x,y\right)=f\left(x\right)+\langle y,Ax-b\rangle +\frac{\rho }{2}{∥Ax-b∥}^{2} \end{array}$$
where $y\in R^{m}$ is the *Lagrangian multiplier*. Then, the ADMM method updates *x* and *y* by optimizing $L_{\rho }\left(x,y\right)$ alternatively. In particular, suppose we are given feasible vectors $\left(x^{k},y^{k}\right)$ at the *k*th iteration. Then,
(8)$$\begin{array}{c} x^{k+1}={\mathrm{argmin}}_{x}\;L_{\rho }\left(x,y\right) \end{array}$$
(9)$$\begin{array}{c} y^{k+1}=y+\rho \left(Ax^{k+1}-b\right) \end{array}$$
where $x^{k+1}$ is a global minimizer of $L_{\rho }\left(x,y^{k}\right)$ for a fixed $y^{k}$, and $y^{k+1}$ is the result of a one-step gradient ascent with step size ρ. Here the step size for $y^{k+1}$ could be different from the penalty parameter ρ so that $y^{k+1}=y^{k}+\alpha \left(Ax^{k+1}-b\right)$ for some α≠ρ. However, it is common to use α=ρ. Then, we obtain a sequence $\left\{x^{k},y^{k}\right\}$ of vectors and [34] shows this sequence converges to a stationary point $\left(x^{*},y^{*}\right)$ that satisfies (Section 3).

The paper [31] adapts the ADMM framework to solve the NMF problem with the Bregman divergence. They firstly reformulate the problem (Equation 2) by introducing additional auxiliary variables as follows:(10)$$\begin{array}{cc} \underset{W,H,X,W_{+},H_{+}}{\mathrm{minimize}}\; & D(V|X) \end{array}$$(11)$$\begin{array}{cc} \;s.t.\;\;\; & X=WH \end{array}$$(12)$$\begin{array}{cc} \; & W=W_{+},H=H_{+} \end{array}$$(13)$$\begin{array}{cc} \; & W_{+}\geq 0,H_{+}\geq 0 \end{array}$$
where *X*, $W_{+}$, and $H_{+}$ are additional auxiliary variables. The corresponding augmented Lagrangian function of (Equation 2) is given by
$$\begin{array}{cc} L\rho (X,W,H,W_{+},H_{+}, & \alpha_{X},\alpha_{W},\alpha_{H})=\\ \; & D\left(V,X\right)+\langle \alpha_{X},X-WH\rangle +\frac{\rho }{2}{∥X-WH∥}_{F}^{2}\\ \; & +\langle \alpha_{W},W-W_{+}\rangle +\frac{\rho }{2}{∥W-W_{+}∥}_{F}^{2}\\ \; & +\langle \alpha_{H},H-H_{+}\rangle +\frac{\rho }{2}{∥H-H_{+}∥}_{F}^{2} \end{array}$$
where $\alpha_{X}$, $\alpha_{W}$, and $\alpha_{H}$ are the Lagrange multipliers. The updates are taken by minimizing $L_{\rho }$ alternatively with respect to each primal variable and taking a gradient ascent in each of the Lagrange multipliers. The algorithm is summarized in Algorithm 1.
**Algorithm 1:** ADMM for NMF [31].
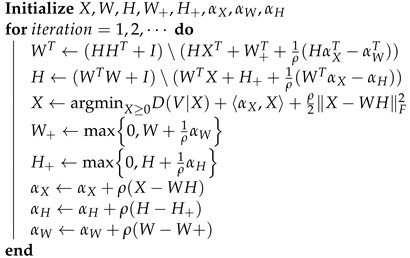


Here, the notation A∖b is an operator in Matlab that takes the inverse of *A* and multiplies it by *b*, that is, $A\setminus b:=A^{-1}b$.

## 4. Block-Active Method

In this section, we consider a constrained optimization problem formulated as follows:(14)$$\begin{array}{c} \min\;f(x)\;\;s.t.\;\;x\geq 0, \end{array}$$
where *f* is a convex objective function, and $x\in R^{n}$ is the decision variable. Section 5 proposes a new ADMM-type method where the problem (Equation 14) is an important subproblem. We propose to use a *block coordinate descent* (BCD) method to solve the problem (Equation 14). In general, a BCD method picks up a block of coordinates of the decision variable and minimizes the objective function only with respect to the selected block of coordinates. In particular, let $x^{k}$ be the current feasible point at the *k*th iteration. Let $i_{k}$ be the selected coordinate. Then, the update rule is given by
$$\begin{array}{c} x^{k+1}=\mathrm{argmin}\;f\left(x+e_{i_{k}}v\right),\;s.t.\;x+e_{i_{k}}v\geq 0 \end{array}$$
where $e_{i_{k}}$ is a vector whose entries are all zeros, except the $i_{k}$th entry is equal to 1. How to select the block is significant in a BCD method. In general, there are three ways to select the block, that is, cyclic, random, and greedy. In the cyclic selection rule, each block is selected cyclically. Each block is selected randomly if the random selection rule is applied. A block is selected if it has the largest magnitude of the partial derivative, that is,
$$\begin{array}{c} i_{k}={\mathrm{argmax}}_{i\in [n]}\left|\frac{\partial f(x^{k})}{\partial x_{i}}\right| \end{array}$$
where [n]={1,2,⋯,n}.

### 4.1. Block-Active Method

Here, we propose a new block coordinate method called *block-active method* to solve the problem (Equation 14) where the block is selected based on the *stationary condition*. Note that a vector $x^{*}$ is a stationary point of (Equation 14) if it satisfies
(15)$$\begin{array}{c} \left\{\begin{array}{cc} \frac{\partial f(x^{*})}{\partial x_{i}}=0, & \mathrm{if}\;x_{i}^{*}>0\\ \frac{\partial f(x^{*})}{\partial x_{i}}\geq 0, & \mathrm{if}\;x_{i}^{*}=0. \end{array}\right. \end{array}$$

Here, Equation (Equation 15) is called *stationary condition*. At each iteration, we collect the coordinates that do not satisfy the stationary condition. In particular, let x≥0 be a feasible point. We construct an index set F as follows:(16)$$\begin{array}{c} F=\left\{i\in \left[n\right]:x_{i}>0\vee \left(x_{i}=0\wedge \partial f\left(x\right)/\partial x_{i}<0\right)\right\}. \end{array}$$

Note that here, we include some extra coordinates in F, that is, $x_{i}>0$ and $\frac{\partial f(x)}{\partial x_{i}}=0$. Later on, we show that if *x* is already a stationary point, including these extra coordinates does not make the block-active method move away from a stationary point. Instead, if *x* is not a stationary point, with the help of a scale matrix *H*, including these extra coordinates make the proposed method converge faster.

Given the index set F, we define vectors *g* and *d* as follows:(17)$$\begin{array}{c} g_{i}:=\frac{\partial f(x)}{\partial x_{i}},\;\forall i\in F\;\mathrm{and}\;d:=-H^{-1}g \end{array}$$
where $g\in R^{\left|F\right|}$, $d\in R^{\left|F\right|}$, and $H\in R^{\left|F|\times |F\right|}$ is a *strictly positive definite (p.d.) matrix*. Given a scalar α>0, we define a single-variable function x(α) as follows:(18)$$\begin{array}{c} x{\left(\alpha \right)}_{i}=\left\{\begin{array}{cc} \max\{0,x_{i}+\alpha d_{i}\}, & i\in F,\\ x_{i}, & i\notin F. \end{array}\right. \end{array}$$

From Equation (Equation 18), we can see only part of vector *x* is selected and updated. The subvector of *x* is selected based on the stationary condition (Equation 16). The algorithm is summarized in Algorithm 2. As noted by [35], our method has the potential to be extended in a distributed manner.
**Algorithm 2:** Block-active method to minimize (Equation 14)
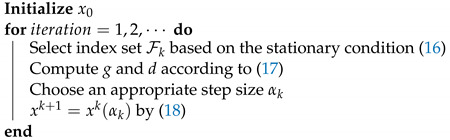


### 4.2. Convergence Analysis of the BCD Method

Given a feasible point *x*, we can show *x* is a stationary point if and only if the single variable function x(α)=x for all strictly positive α>0. On the other hand, if *x* is not a stationary point, then we can show there exists a strictly positive scalar $\bar{\alpha }$ for which any $\alpha \leq \bar{\alpha }$ causes a descent in the objective value.

**Theorem 1.** 
 *(1)* 
*x≥0 is a stationary point of (Equation 14) if and only if x=x(α) for all α>0.*
 *(2)* 
*If x is not a stationary point, then there exists $\bar{\alpha }>0$ such that*

(19)
$$\begin{array}{c} f(x(\alpha ))<f(x),\;\;\mathrm{for}\;\mathrm{all}\;0<\alpha \leq \bar{\alpha }. \end{array}$$




**Definition 1.** 
*A function g is called L-smooth if ∇g is Lipschitz continuous with constant L>0. In particular, there exists a strictly positive scalar L>0 for which*

(20)
$$\begin{array}{c} ∥\nabla g(x)-\nabla g(y)∥\leq L∥x-y∥,\;\mathrm{for}\;\mathrm{all}\;x,y. \end{array}$$



Given a function *g* is *L*-smooth, we have the following well-known descent lemma.

**Lemma 1** (Descent Lemma)**.** *If g is L-smooth, then*
(21)$$\begin{array}{c} g(y)\leq g(x)+\langle \nabla g(x),y-x\rangle ,\;\mathrm{for}\;\mathrm{all}\;x,y. \end{array}$$

Suppose the objective function *f* in (Equation 14) is *L*-smooth. It follows from the descent lemma that the sequence $\left\{f\left(x^{k}\right)\right\}$ generated by the *block-active method* consistently decreases and it converges to $f(x^{*})$ sublinearly.

**Theorem 2** (Convergence result)**.** *Assume the objective function f is L-smooth and $\lambda_{\min}\left\{Q^{k}\right\}\geq \mu$ for all k, where μ>0 is a fixed constant. If $\alpha_{k}\leq \min\left\{{\bar{\alpha }}^{k},\mu^{2}/L\right\}$, where ${\bar{\alpha }}^{k}$ is defined in Theorem 1, then*
(22)$$\begin{array}{c} f\left(x^{k}\right)-f\left(x^{*}\right)\leq \frac{∥x^{0}-x^{*}{∥}_{Q}^{2}}{k}, \end{array}$$
*where ${∥z∥}_{H}^{2}:=\sum_{i,j\in F}z_{i}Q_{ij}z_{j}$.*

In the above theorem, we demonstrate that the error terms defined as $f\left(x^{k}\right)-f\left(x^{*}\right)$ are consistently diminished. This reduction is characterized by the relation $f\left(x^{k}\right)-f\left(x^{*}\right)=O\left(k^{-1}\right)$, indicating the gradual decline to zero as the iteration count *k* approaches *∞*. This type of convergence behavior is so-called sublinear [22]. It stands in contrast to the linear convergence rate in the form of $\gamma^{k}$ for some constant γ∈(0,1), where the reduction is constant. On the other hand, the smoothness assumption of the objective function *L* can be dropped but the convergence can still be ensured by using the arguments introduced in [36,37] for the non-Lipschitz optimization. The proof of Theorem 1 and 2 can be found within Appendix A and Appendix B.

## 5. Block-Active ADMM

In this section, we propose a new ADMM-type method to solve the NMF problem (Equation 2) by using the *block-active method* to solve the subproblems. Particularly, since the intermediate quantity X=WH needs to be updated repeatedly once the matrices *W* and *H* are updated, we directly introduce this quantity as a new variable in the optimization problem. Thus, the NMF problem becomes
(23)$$\begin{array}{cc} \min\; & D(V|X),\\ s.t.\; & X=WH,\\ \; & H,W\geq 0. \end{array}$$

Since the ADMM framework is good at dealing with equality constraints, we propose a new algorithm based on the ADMM framework by introducing one dual variable $\alpha_{X}$. The corresponding augmented Lagrange function is given by
(24)$$\begin{array}{cc} L_{\rho }\left(X,W,H,\alpha_{X}\right)= & D\left(V|X\right)+\langle \alpha_{X},X-WH\rangle +\frac{\rho }{2}||X-{WH||}_{F}^{2}. \end{array}$$

The updates alternately optimize $L_{\rho }$ with respect to each of the three primal variables, followed by one update on the dual variable. The updates are summarized as follows.
(25)$$\begin{array}{cc} W^{+} & ={\mathrm{argmin}}_{W\geq 0}L_{\rho }\left(X,W,H,\alpha_{X}\right) \end{array}$$
(26)$$\begin{array}{cc} H^{+} & ={\mathrm{argmin}}_{H\geq 0}L_{\rho }\left(X,W^{+},H,\alpha_{X}\right) \end{array}$$
(27)$$\begin{array}{cc} X^{+} & =\mathrm{argmin}L_{\rho }\left(X,W^{+},H^{+},\alpha_{X}\right) \end{array}$$
(28)$$\begin{array}{cc} \alpha_{X}^{+} & =\alpha_{X}+\rho \left(X^{+}-W^{+}H^{+}\right) \end{array}$$

Since the optimization with respect to *X* does not have any constraint, $X^{+}$ has a closed-form solution by solving the equation $\frac{\partial L_{\rho }}{\partial X}=0$. The closed-form solution is given in ([31], Theorems 1 and 2). In contrast, the updates for *W* and *H* can be reformulated in the form of *nonnegative least squares*. Taking the optimization of *H* as an example, we have
$$\begin{array}{cc} H^{+} & ={\mathrm{argmin}}_{H\geq 0}L_{\rho }\left(X,W^{+},H,\alpha_{X}\right)\\ \; & ={\mathrm{argmin}}_{H\geq 0}\langle \alpha_{X},X-WH\rangle +\frac{\rho }{2}||X-{WH||}_{F}^{2}\\ \; & ={\mathrm{argmin}}_{H\geq 0}\frac{\rho }{2}\left[\frac{1}{\rho^{2}}∥\alpha_{X}{∥}^{2}+2\langle \alpha_{X}/\rho ,X-WH\rangle +||X-{WH||}_{F}^{2}\right]\\ \; & ={\mathrm{argmin}}_{H\geq 0}\frac{\rho }{2}||X-WH+\alpha_{X}/\rho {\left|\right|}_{F}^{2}\\ \; & ={\mathrm{argmin}}_{H\geq 0}\left|\right|\left(X+\alpha_{X}/\rho \right)-WH{\left|\right|}_{F}^{2} \end{array}$$

Thus, this subproblem can be solved by the method we proposed in the previous section, called *block-active method*. In particular, we can choose the scaling matrix *Q* as part of $W^{T}W$ based on the index set F. Since $W\in R^{M\times K}$ and M≫K, $W^{T}W$ is highly likely strictly positive definite so that *Q* is a submatrix of $W^{T}W$. Moreover, we denote nnls_blockactive(A,B) as the procedure proposed in the previous section and used to solve the nonnegative constraint problem in the form of
(29)$$\begin{array}{c} {\min∥AX-B∥}^{2}\;s.t.\;X\geq 0. \end{array}$$

Algorithm 3 is provided as an example of the proposed block-active ADMM method for the case where β=1 in the β-divergence distance.
**Algorithm 3:** Block active ADMM.
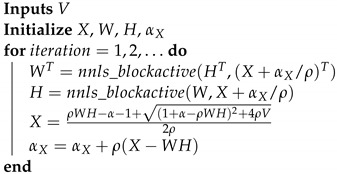


**Remark 1.** 
*As established in ([32], Theorem 2), the alternating direction method of multipliers (ADMM) demonstrates convergence to a stationary point, when descents are achieved for subproblems within each iteration, despite the global problem being nonconvex. In our context, we are addressing a nonconvex optimization challenge, specifically the nonnegative matrix factorization (NMF) problem as defined in Equation (Equation 2) and subsequently reformulated in Equation (Equation 23). Through the utilization of the ADMM framework, we strategically partition the problem into a series of subproblems, each solvable within an iteration. Leveraging the convergence assurance provided by Theorem 2, we can confidently assert that a descent is guaranteed within each subproblem. Consequently, invoking the findings of ([32], Theorem 2) within our specific context, we secure a robust convergence result for the proposed method delineated in Algorithm 3, leading to the attainment of a stationary point.*


## 6. Numerical Experiments

### 6.1. Synthetic Datasets

We first tested the proposed algorithm on a moderately synthetic dataset with m=500, n=500, and k=150. We generated the ground truth $W_{0}$ and $H_{0}$, and $V=W_{0}H_{0}$. We examined the performance of the proposed algorithm against the standard multiplicative update [38] and the ADMM [34]. We set ρ=1 and the maximum iteration to be 1000. The performance results are shown in Figure 3. We can see that the proposed block method can achieve a much lower error level given the same amount of time.

### 6.2. Real Datasets

We evaluated the proposed method against both the multiplicative update and ADMM algorithms using real datasets. These datasets were generated using either a 2D imaging sensor or a near-infrared (NIR) imaging sensor.

*UMist* (https://cs.nyu.edu/~roweis/data.html, accessed on 2 January 2022): This dataset is an image dataset containing 575 images of 20 people, which consist of images of individuals captured in various poses, ranging from profile to frontal views. All files in the dataset are in the PGM format, have a resolution of approximately 220×220 pixels, and are 256-bit grayscale images.*ORL* (http://www.cad.zju.edu.cn/home/dengcai/Data/FaceData.html, accessed on 2 January 2022): The dataset was generated by a 2D imaging sensor and includes 400 different images of each of 40 distinct individuals, where each image has 92×112 pixels and a depth of 256 levels of gray per pixel. The photographs were taken on different occasions, with variations in lighting, facial expressions, and facial features.*COIL* (http://www.cad.zju.edu.cn/home/dengcai/Data/MLData.html, accessed on 2 January 2022): The dataset contains 7200 images in the form of 32×32 pixels for 100 objects. The images were captured on a motorized turntable against a black background. The dataset was utilized in a real-time recognition system that employed a sensor to detect the objects and display their angular pose.*YaleB* (http://www.cad.zju.edu.cn/home/dengcai/Data/FaceData.html, accessed on 10 February 2023): The dataset consists of image data generated by a 2D imaging sensor. It comprises 2414 images of size 192 × 168 pixels from 38 individuals. The images were taken under different lighting conditions and a variety of facial expressions.*NIR* (http://vcipl-okstate.org/pbvs/bench/Data/07/download.html, accessed on 10 February 2023): The dataset was created via a near-infrared (NIR) imaging sensor. It includes 3940 NIR face images of 197 persons. The images have a size of 480×640 pixels, 8-bit, and are not compressed.

The results are shown in Figure 4, Figure 5, Figure 6, Figure 7 and Figure 8. Based on the results on the real datasets, we can see the objective value using multiplicative update decreases faster at the beginning, but later, the proposed block method can converge to a better solution which has a much lower error level. In addition, comparing to the ADMM, the proposed block method is much more stable. In Figure 7, the objective value using the ADMM does not consistently decrease. That is because ρ=1 is too small for the YaleB dataset. However, using ρ=1, the proposed block method does not diverge, and the objective value continuously decreases. The source code for the proposed algorithms has been added to GitHub, accessible at https://github.com/Xinyao90/Block-active-ADMM-to-Minimize-NMF-with-Bregman-Divergences.git, accessed on 10 August 2023.

## 7. Conclusions

In this paper, we proposed a new block method that aimed to solve the nonnegative matrix factorization problem using the general Bregman divergence distance metric. Nonnegative matrix factorization is a widely used technique in various fields such as image processing, speech analysis, and bioinformatics. In particular, image processing systems heavily rely on databases generated by existing imaging sensors, and the efficiency and accuracy of these systems depend on the performance of the nonnegative matrix factorization method used to process these databases.

Our proposed block method was built on the framework of the alternating direction method of multipliers (ADMM), which is a popular algorithm used to solve optimization problems. However, instead of following the traditional approach of the ADMM to solve the subproblems, we introduced a new method that employed a block coordinate method. In this approach, we selected a block based on the stationary condition, which allowed us to converge faster and to a solution with a lower error level compared to the previous ADMM method.

To demonstrate the effectiveness of our proposed method, we conducted a series of numerical experiments. The experiments included comparisons of our block method with the traditional ADMM method and other state-of-the-art methods in terms of runtime and overall accuracy. Our numerical results showed the dominance of our proposed block method over other methods, highlighting its effectiveness in solving the nonnegative matrix factorization problem using the general Bregman divergence distance metric.

In summary, our proposed block method provides an efficient and accurate solution to the nonnegative matrix factorization problem using the general Bregman divergence distance metric. Its unique approach to solving subproblems using a block coordinate method has proven to be faster and more accurate than traditional methods, as demonstrated by our numerical experiments. Our proposed method can help improve the performance of image processing systems and other applications that rely on nonnegative matrix factorization.

## Figures and Tables

**Figure 2 sensors-23-07229-f002:**
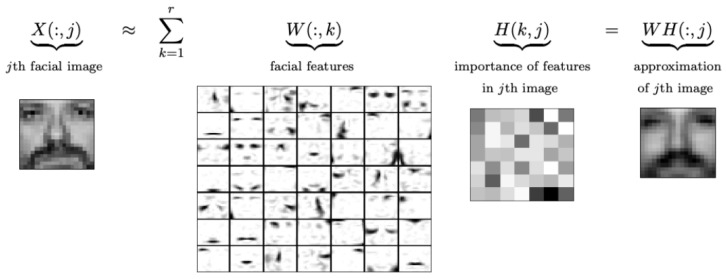
NMF face recognition [21].

**Figure 3 sensors-23-07229-f003:**
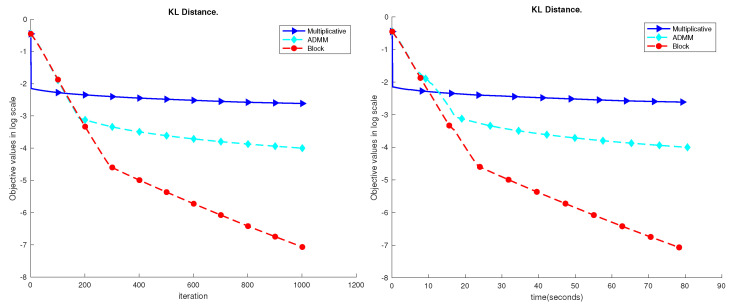
Performance comparison of the synthetic dataset. Here, we set m=n=500 and k=150. The maximum iteration is set to be 1000. We record the objective value on the log scale. We can see the proposed block method can achieve a much lower error level given the same amount of running time. In another word, the block method is faster to achieve the specified accuracy than the other two methods.

**Figure 4 sensors-23-07229-f004:**
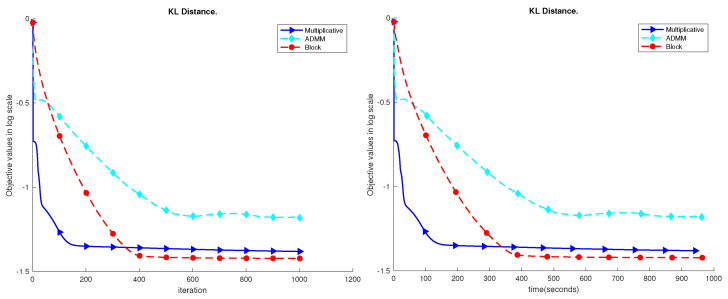
Performance comparison of the UMist dataset.

**Figure 5 sensors-23-07229-f005:**
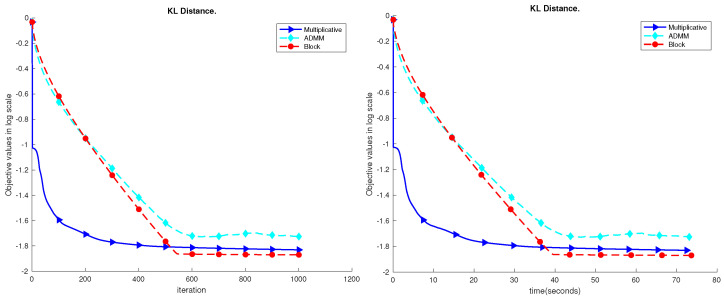
Performance comparison of the ORL dataset.

**Figure 6 sensors-23-07229-f006:**
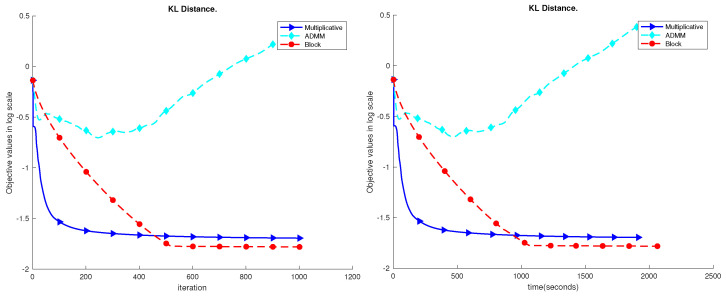
Performance comparison of the COIL dataset.

**Figure 7 sensors-23-07229-f007:**
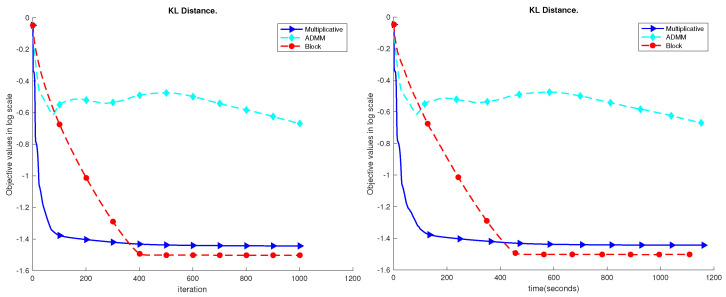
Performance comparison of the YaleB dataset.

**Figure 8 sensors-23-07229-f008:**
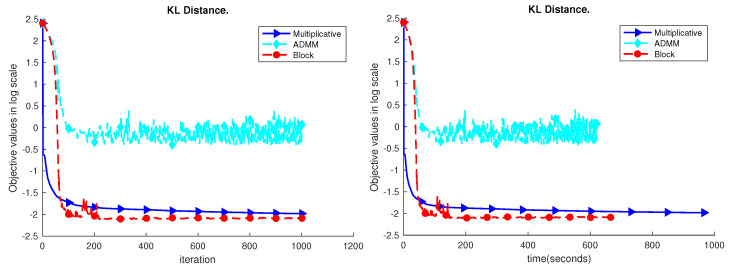
Performance comparison of the NIR dataset.

## Data Availability

Not applicable.

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
