# Peer review of "Block-Active ADMM to Minimize NMF with Bregman Divergences"

_sensors, 2023, doi:10.3390/s23167229_

Round 1
Reviewer 1 Report
This paper proposes a new Alternating Director Method of Multipliers (ADMM) to solve the NMF problem with general Bregman divergence by using the method of block active. Theoretical analysis and experimental results show that the proposed algorithm converges faster than the previously proposed algorithms.
Overall, the proposed method is innovative and the experiments are sufficient. However, there are some minor concerns to be addressed.
1. Compared with the same type of methods, whether the proposed method will achieve better results, that is, the function ||X -W*H*||_F^2 can have a smaller value for the proposed method.
2. What are the advantages of the proposed method as compared with available ADMM methods?
3. References to the multiplicative update and ADMM methods need to be provided in the experimental part.
The language needs to be polished, and then the readability can be improved.
Reviewer 2 Report
The paper proposes a block active ADMM method to minimize the NFM problem with general Bregman divergences. A BCD-type method is used for the subproblems of ADMM, where each block is chosen directly based on the stationary condition. As a contribution of the work, the method converges faster than other algorithms and uses much fewer auxiliary variables.
The paper is well-structured and presents results interesting to readers in ML from both computational and mathematical perspectives.
Minor issues:
Line 109 - ...that satisfies(??)
Algorithm 3 - Where is W^T used in the pseudocode after the first line? there - H=nnls_blockactive(W^T,....)?
Adding a link reference to a repository (GitHub) where the source code can be downloaded is suggested.
Reviewer 3 Report
The following aspects must be clearly addressed:
-specify more clearly what are the novelties of the proposed method compared with other existing ones
-the discussion section must be more elaborated: 1) in abstract it is the affirmation: "use much fewer auxiliary variables" - please explain this more clearly in discussion section, 2) in abstract it is the affirmation: " the proposed algorithm is proved to converge to a stationary point sublinearly" - please explain this aspect more clearly in discussion section 3) are there any differences depending on the task that will be solved, 4) the image resolution can influence the result?
-figures must be placed after their first appearance in the text
Round 2
Reviewer 3 Report
Since all my comments were addressed I recommend to publish the paper.